Paediatrics
Open

# Understanding parents' experiences of care for children with medical complexity in England: a qualitative study

Emma Victoria McLorie ,[1] Julia Hackett,[1] Lorna Katharine Fraser[2]

[1]Department of Health Sciences, University of York, York, UK
[2]Cicely Saunders Institute, Department of Women and Children's Health, King's College London, London, UK

**Correspondence to**
Emma Victoria McLorie; emma.mclorie@york.ac.uk

## ABSTRACT

**Objectives** To understand parents' experiences of receiving care for their child with medical complexity.
**Design** Qualitative semi-structured interviews with parents of children with medical complexities across England analysed using reflexive thematic analysis.
**Results** Twenty parents from four hospital sites in England took part in the study, a total of 17 interviews were completed, 3 joint and 14 single parent interviews. Four themes were developed: parents feel abandoned; continuity of care (made up of three subthemes); equipment barriers; and charities fill the gaps.
**Conclusions** The perceived quality of healthcare provided to this population was found to be inconsistent, positive examples referred to continuity of care; communication, clinical management and long-lasting relationships. However, many experienced challenges when receiving care for their children; many of these challenges have been highlighted over the past two to three decades but despite the children's needs becoming more complex, little progress appears to have been made. Parents were seen as adopting significant additional roles beyond being a parent, but they still find themselves left without support across all areas. These families require more structured support. Policy makers and commissioners need to prioritise the needs of families to enable health and social care services to provide the support required.

## BACKGROUND

Children with medical complexity (CMC) have medical fragility and multiple care needs that are not easily met by existing healthcare models. They can be defined as 'children with characteristic patterns of needs, chronic conditions, functional limitations, and healthcare use'.[1] However, there are many other terms used, some similar, with overlapping populations of children, for example, complex chronic conditions,[2] life-limiting or life-threatening conditions[3] and medical fragility[4]. Due to advances in medicine and use of technologies; there are more CMC living longer.[5] Much of the existing literature on models of care for this population has

## WHAT IS ALREADY KNOWN ON THIS TOPIC

⇒ Children with medical complexity are a growing population with increasing levels of needs. There is a lack of contemporary information about parental experiences in the UK.

## WHAT THIS STUDY ADDS

⇒ Parents of children with medical complexity experience challenges when interacting with health services such as poor communication and coordination. Parents found themselves left without support as extensive gaps in relation to available psychological and financial support were made visible. Parents were dependent on charities for essential equipment, respite and financial support.

## HOW THIS STUDY MIGHT AFFECT RESEARCH, PRACTICE OR POLICY

⇒ The challenges experienced by parents of children with medical complexity when receiving care for their children suggest that changes to care and support are required. Although some existing services were positively received, this was fragmented and not available to all parents. Therefore, further expertise in the community and hospital sites may be beneficial. The level of dedicated support available was limited, meaning that psychological support should be made essential. Parents also faced difficulties when accessing charity support, indicating that changes to the application process by policy makers is needed.

been undertaken in North America where healthcare is funded and delivered in a very different way in comparison to the UK's free at point of care, publicly funded health system. Therefore, the generalisability of US research to the UK population has been questioned.[6] A previous UK-based report provided guidance on transitional care from the perspective of children using long-term ventilation, illustrating their complex needs[7], recent reviews of critical care services for children by NHS England and Improvement[8] highlighted the growing population of children with complex

underlying medical conditions, but did not make any specific recommendations about their care. However, the need to redesign health services for children was emphasised in the NHS Long Term Plan[9] and in other reports.[10]

Complexities surrounding their child's condition are known to impact parents' psychological well-being.[11] In the USA, research has shown that parental well-being is further impacted when interacting with health services as issues such as poor communication have been historically reported.[12] In other studies, examining clinician and parental perspectives when accessing emergency care,[13] it was found that professionals did not communicate with each other, leaving parents to take control and facilitate communication, illustrating an existing problem. To reduce implications, complex services across the USA and North America have been implemented.[14] In a previous literature review examining the reported costs of this population,[15] it was found that greater emphasis on the family dynamic needed to be considered by policymakers and future research. Suggestions have also been made to gather UK-based families' experiences to determine the most effective model.[16] To address such recommendations, this study aimed to understand parents' instances of receiving care for their child with medical complexities across England.

## METHODS
### Design
An exploratory phenomenological qualitative study using semi-structured interviews to understand parents of CMC accounts and experiences of receiving care for their child.

### Patient and public involvement
No patients were involved in this study. However, the study design, initial findings and outputs were discussed in patient and public involvement (PPI) meetings. These meetings involved members of the Martin House Research Centre Family Advisory Board[17] made up of parents of young people with complex healthcare needs and life-limiting conditions. Members also gave their input of the semi-structured interview topic guide (see online supplemental file 1) and EVM conducted a pilot interview with one member. Findings will be disseminated in a series of presentations in key meetings, reports and publications.

### Participants
Purposeful sampling was used across four tertiary hospitals in England between March and September 2022, three of which had formal services for CMC and one that did not have a formalised service. These hospital sites were chosen as each were known to treat CMC. The three hospitals which did have existing services also differed from one another, some with more established teams whereas others were relatively new and involved a single individual. Therefore, it was likely that experiences may

**Table 1** Inclusion and exclusion criteria

| Inclusion criteria | Exclusion criteria |
| --- | --- |
| Parent is 16 years old or older | Parent is under 16 years old |
| Their child has medically complex needs | Their child does not have medically complex needs |
| Their child is 19 years old or younger | Their child is older than 19 years old |
| Those who can provide capacity to participate in the study, guided by the 2005 Mental Capacity Act | Those who lack capacity to participate in the study, guided by the 2005 Mental Capacity Act |

Ethical and governance approvals were obtained from HRA and Health and Care Research Wales (08/09/21, 300516) and Research Ethics Committee (08/09/21, 21/WA/0257).

differ depending on type of service received. For eligibility criteria, see table 1.

### Recruitment
Parents were recruited through clinicians located at the four NHS sites or via social media (Twitter). Clinicians provided parents with relevant study information via post, telephone or during consultations. If interested, parents received consent to contact form to return to the research team. Once received by the study team, parents were contacted to check eligibility and begin the interview set-up process. If contacted via social media, parents were contacted directly by the research team using the parents preferred contact method, via e-mail or telephone. Participants were recruited between March and September 2022.

### Data collection
The interview topic guide (see online supplemental file 1) was developed, piloted and discussed in PPI meetings with the Martin House Research Centre Family Advisory Board, which consists of parents and family members of children and young people with life-limiting conditions and complex healthcare needs.[17]

According to parents' preference, telephone or video-call interviews were carried out in a private office by a female researcher (EVM), experienced in interviewing and not previously known to parents. Prior to the interview, informed consent for participation and future publications was obtained. Probes and follow-up questions were used to better our understanding. During the interview, field notes were taken to reduce researcher bias, noting impressions and observations which later assisted during the analysis process. After interviews were conducted by EVM, debriefs were held with LF or JH, which also allowed opportunity to discuss any potential thoughts or potential bias. All interviews were recorded and later transcribed verbatim. Transcripts were not returned to participants due to time constraints.

## Data analysis

Data was analysed using a reflexive thematic analysis approach,[18] an accessible framework which is useful in capturing parallels and contrasts among varied perspectives.

The analysis process involved a total of six steps; (1) anonymised interview transcripts were read and re-read to begin the familiarisation process, initial thoughts and impressions were noted by EVM; (2) after familiarising oneself with the data, EVM uploaded transcripts and coded inductively using NVivo12; (3) throughout the analysis process, team members (LF, JH and EVM) would frequently meet to discuss initial codes, exploring differing opinions, input and to limit bias; (4) codes were then sorted into initial themes by examining potential patterns and shared characteristics; (5) themes were reviewed among team members and later defined and named; (6) themes were finalised and later written up. In relation to data saturation and aligning with Braun and Clarke's theoretical guidance[19] when using reflexive thematic analysis, we adopted the practice known as information power.[20] Information power refers to five dimensions: (1) study aim, (2) sample specificity, (3) use of established theory, (4) quality of dialogue and (5) analysis. Our aim in this study was closely defined meaning that participant characteristics and theoretical basis were too. The in-depth interview contained a strong dialogue between the researcher and the participant followed by an in-depth analysis. Both of which assisted in accomplishing information power. It is, therefore, believed that our sample size enabled us to achieve information power.

**Table 2** Child characteristics

| Parent and child characteristics | N | |
|---|---|---|
| Number of parents | 20 | |
| Type of parent | | |
| | 14 | Mum |
| | 6 | Dad |
| Number of children | 18 | (One set of siblings, twins) |
| Sex of child | | |
| | 4 | Girl |
| | 14 | Boy |
| Age range of child | | |
| | 10 | Aged 0–5 |
| | 4 | Aged 6–10 |
| | 3 | Aged 11–15 |
| | 1 | Aged 16–20 |
| Child's diagnosis | | |
| | 8 | Neurological condition |
| | 6 | Genetic condition |
| | 4 | Congenital condition |
| Age at diagnosis | | |
| | 3 | At birth |
| | 11 | Infancy (0–1 years) |
| | 3 | Childhood (1–9 years) |
| | 1 | Unknown |

## RESULTS

Twenty parents with CMC took part in interviews, 3 of which were joint interviews with both parents, making it a total of 18 interviews from 3 tertiary hospital sites located in England (one site failed to recruit any participants). Four parents (one couple, one father and one mother) were also recruited, but declined to take part in an interview for unknown reasons. Mean interview length was approximately 60 min (range: 56–120 min). Parents were not asked to provide feedback on the findings. However, EVM presented findings to members of the Martin House Research Family Advisory Board. Table 2 provides an overview of the parent and child characteristics.

Four analytical themes were developed during the analyses, one included three subthemes as further illustrated in table 3 below.

## Theme 1: parents feel abandoned

Discussing their experiences, parents often reflected on the time in which they received their child's diagnosis. It was felt that there was a limited amount of available support, many were simply provided with the name of the medical condition and felt abandoned. To receive information about their child's condition, some relied on internet searches.

They just give you names and then walk out the room and think you'll be happy for being given a name. (P17)

You're not getting enough support and information then you have no choice and when I googled that … I really broke down. (P22)

In one instance, a couple were told that reasoning for the limited emotional support provided by clinicians was due to their training focusing on the medical issue and not the emotional impact on families.

She said, "Doctors are trained, and they go through the medical evidence, and they don't look at how the parent might be feeling at that time" [referring to a statement made by their consultant]. (P03&04)

As parents moved beyond the diagnosis period and through the healthcare setting, the limited support and understanding of their experiences appeared to continue, one couple described it as being 'abandoned' (P14&15). In addition to limited support surrounding their child's diagnosis, parents expressed feeling as

**Table 3** Analytical themes

| Analytical themes | Description of themes |
|---|---|
| Theme 1: parents feel abandoned | Parents did not always feel emotionally supported beginning from the diagnosis period continuing as they move through the health system. Parents found themselves isolated and fighting for coordination of their child's care. Parents made suggestions for future services to provide emotional and physical support for parents. |
| Theme 2: continuity of care | Parents described both positive and negative experiences relating to continuity of care. Parents experienced three types of continuity: information, management and relationship. Each were categorised into three subthemes. |
| Subthemes:<br>1. Information continuity<br>2. Management continuity<br>3. Relationship continuity | 1. Referred to parents experiencing positive examples of communication between professionals, patients and themselves. It also pinpoints some poor examples of information sharing among professionals and parents.<br>2. Referred to positive examples of when services assisted in coordination and clinical management. It also explored some examples of when parents experienced issues surrounding inappropriate hospital settings such as A&E.<br>3. Referred to parents appreciating long-lasting relationships with clinicians and other professionals. In instances of when there was a lack of relationship continuity, for example, unknown to staff, implications were found. |
| Theme 3: equipment barriers | Parents found themselves facing barriers when ordering equipment and medication including lengthy waiting times and unsuitable NHS-funded equipment. Assistance from the government mobility service was also not accessible for this population. |
| Theme 4: charities fill the gaps | Parents felt as though there was a lack of financial support available, leaving many to rely on charities to provide them with equipment and respite. |

though their concerns were not listened to. Ultimately, feeling as though their voice was lost.

> I expressed my frustration to every single professional that I was speaking to. Sometimes I felt as though my voice was just bouncing off bare walls and hitting me back (P19)

Feelings of being ignored were also felt when parents tried to communicate the health concerns of their child. However, this was dismissed, illustrating other ways in which parents would feel unsupported despite feeling as though they knew their child best.

> "This is not normal behaviour for him. No, we are not medical professionals… But we are telling you it is definitely not this, or it is definitely not that" That is ignored every single time [describing communication with professionals] (P14&15)

All of which appeared to have a negative effect on parents, many feeling as though they were pushed to their limits, experiencing feelings of exhaustion.

> You were just basically left to carry on until you couldn't. (P17)

Feelings of despair were worsened due to staff shortages with one couple unable to leave the hospital site as ward staff were unavailable to provide one-to-one care. This meant that the couple had to perform care duties

instead for an excessive amount of time, describing feeling as though they were at 'breaking point' (P10&11). In attempts to receive support and inclusion in their child's care, parents expressed feeling as though their 'voice is lost' (P20) when interacting with health and social services.

Alternatively, parents relied on guidance from other families and would attempt to support each other. In one example, a parent describes their interaction with another parent who had also received limited support from hospital services.

> I met a woman with a son and he'd been in hospital for 7 months, she didn't speak very good English, nobody had offered her a translator … [support team name] I phoned them up and said could they bring her application for DLA and they said, "No, she's not entitled to DLA". She is entitled to DLA …. She's got a translator now, she's applied for DLA now, but only because like me and one of the other mums pushed it, but nobody's volunteering that information to her. (P24)

Alongside relying on other families, parents also found themselves with little choice but to become coordinators in their child's care, describing it as a 'fight' (P22). Parents were thought to navigate many aspects of their child's care, undertaking many roles such as an 'advocate' (P22). The coordination role involved parents taking control, arranging and communicating between health and social care professionals.

I'm always having to do chasing and arrange things and tell people this is what this person said (P26)

Due to the experienced isolation and limited support available, parents could not always adopt a parental role or have time for themselves. All of which was further restricted due to their ongoing caregiving duties.

I've now got to be a personal coordinator to book all these people in at various times … Where's the time for me to be mum? (P18)

Ultimately, parents advocated for both physical and emotional support services, which would allow them to have an opportunity to be a parent and lessen the emotional and physical difficulties faced.

Counselling at diagnosis is a huge problem, that should really be offered (P24)

## Theme 2: experiences of care: continuity of care

Describing their experiences of interacting with health and social services, families highlighted both negative and positive experiences, the majority relating to continuity of care. The concept of continuity of care refers to three different aspects: information, management and relationship, which have been divided into three subthemes.

### Subtheme 1: information continuity

Parents expressed appreciation of receiving information continuity between themselves and professionals, valuing consistent communication and being kept informed, described as being 'up to date' and were able to reach out if they were 'worried about anything'. Due to information being effectively shared between professionals and families, the paediatric team made one individual feel as though they were not 'alone' (P19). Another example of meaningful communication and information sharing related to staff members involving parents in the decision-making.

We're not felt like we're rushed to get in and get out … She came in and had a chat with us and said "would this medication be any good for him?"(P08)

In instances of when professionals made the decision to include parents in their child's care, treating them as though they were a 'guide', parents felt 'listened to' (P14&15). Parents also placed importance of when professionals would directly communicate with their child, the patient. Reflecting on an experience with a clinician and their child, one parent describes a meaningful approach used, 'He said, "I've wanted to speak to [child's name] just to make sure that I wasn't leaving him out of the conversation, can he understand me?' (P08). Another type of communication valued was between

professionals, as recent changes made information sharing across sites possible, a smoother way of communicating, 'instead of having this trolley full of a big file of paper' (P12).

Despite many occasions of effective information sharing, there were also many instances of poor communication. In some instances, wards were described as having 'no communication at all', not an isolated incident as 'everyone raises it on a daily basis' (P17). Another reason behind limited communication was thought to relate to a clinician's caseload, as parents could not 'get any response' (P14&15), lessening information continuity. Similarly, when communicating with patients, one parent admitted that some may 'just ignore him completely' (P24). Another factor disrupting information continuity was communication between staff as many parents were left to coordinate between professionals.

We just have a major problem with communication between [hospital name] and [hospital name]. It tends to be me passing on messages … they're not allowed to email between Trusts (P24)

Ultimately, it was agreed that communication was an area that needed to 'improve' (P01) to ensure continuity.

### Subtheme 2: management continuity

Parents welcomed management continuity as it brought feelings of security, knowing that their child's care was being effectively managed. Specialist hospital sites were seen as one of the main contributors as it was believed they were best to manage complex conditions,

[specialist hospital name] is a bit better equipped to deal with children with more additional needs. (P20)

Another benefit of using a specialist hospital site related to parents being familiar with members of staff, meaning that their child's care was coordinated more smoothly,

We go into the A&E … As soon as we get there, they know who we are (P02)

In some hospital sites, dedicated services for CMC had been introduced and would assist with care coordination, something that parents valued, one parent describing it as a faultless service,

The team and the discharge was done beautifully. I can't fault it (P19)

Prior to discharge, many experienced a change in family circumstances due to reduced working hours. Therefore, quite a high number of parents would need to complete application forms for additional financial help. Many of the complex services designed for this population within the hospital sites would assist with 'paperwork' (P03&04) to ensure management of care. Outside

of the hospital site, parents also spoke highly of community nursing services, describing them as 'fantastic' (P19) and 'amazing' (P02) due to effectively managing their child's care, offering support and were available via telephone if required.

Unfortunately, experiences of management continuity were impacted by factors such as the environment, safety concerns, staff shortages and staff expertise. Due to the complex needs of this population, general practice surgeries often did not feel confident in managing their condition and instead, directed parents to the emergency department, an environment which parents deemed as unsuitable.

> They see the ventilator and they're like no, you need to go to A&E and then it's like well what do they even come here for? (P02)

Parents faced implications as they would sometimes wait up to 'six to eight hours just to get seen' (P17), disrupting management of care. Therefore, it was preferred that more community resources were readily available as described below,

> There should be like a 24-hour service where there's somebody on the other end what can tell you what to do, advise you. (P02)

The aftermath of COVID-19 was believed to have impacted community services as there were low staff levels, often meaning that care packages were delayed and would prevent hospital discharge,

> For the carers recruitment is slow and I can only imagine COVID, and Brexit have had a significant impact on recruitment. (P20)

In other instances, hospital sites also experienced staff shortages, which had a negative impact on their care. Due to the complex needs of one child, 24-hour one-to-one care was required. However, it was thought that due to staff shortages and with the parents remaining on the ward, staff would see this patient last, leaving parents to stay late and become tired.

> 24-hour one-to-one … The ward is getting paid for but because we're parents who are there all of the time when they're short-staffed, they staff [child name] last because they know we're more than likely to be there. (P10&11)

The same parents were further impacted by staff shortages when leaving the hospital with their child on a day-release as although they had returned home for much needed rest, it was not always possible as the ward would request their assistance due to being too short-staffed to perform 24-hour one-to-one care on their return,

> We've got back quite late at night 7, 8 o'clock to settle [child's name] so we can get some rest … Before we've got back to the hospital, we've had phone calls asking us would we like stay another night because they're short-staffed. (P10&11)

Despite instances of high confidence levels due to the specialist knowledge, there were safety concerns and inconsistencies in care due to a series of medical errors,

> Not just wearing the same gloves that you've had on with all the patients then you're passing all the germs on to you. Even like touching lines and then you get sepsis … It's just lazy, poor practice. (P17)

There were other safety concerns as some parents did not feel confident leaving their child under the care of the hospital due to staff shortages,

> I don't like leaving him … If they're busy doing another patient and if there's only three members of staff, then there's just not enough staff. (P23)

To help improve management of care and produce continuity across sites, parents made suggestions of an increase in medical complexity professionals and services such as transition.

> More local hospitals need more trained people [when asked what services they would like]. (P02)

### Subtheme 3: relationship continuity

Parents valued long-lasting relationships with professionals, which had been built up over time, creating comfort and relationship continuity. Discussing their relationships with clinicians, parents made extensive efforts to not lose these connections,

> I didn't want to lose Dr. [name]. It was me that insisted that we kept him. (P13)

> He [consultant] has been with us since before [child's name] was born, so we were quite keen that we didn't lose that connection. (P03&04)

In addition to the length of time parents and their child were known to clinicians, active listening and inclusion were contributors to providing positive relationship continuity. For many, it was their first experience of being heard, provoking an emotional and grateful response,

> We've got a really strong relationship with him where we feel like we're listened to as parents. (P10&11)

> He was just so lovely. I cried when we left him because it was the first time anyone had listened to us, paid attention to what we were saying. (P13)

However, it was not common practice for all parents to have one individual involved in their child's care or a relationship that was built up over time, some found themselves experiencing poor continuity as there was not an individual solely involved in their child's care.

It's the lack of continuity and it's the lack of that specific person who could be there if he's admitted that could come and that can bring all the care together because the constant repetition. (P26)

Implications included parents having to repeat information, leaving parents to advocate for a singular clinician,

A single lead consultant [when asked what would be beneficial] (P14&15)

Outside of the hospital setting, general practitioners (GPs) were also a point of continuity in their child's care as over time, relationships were strengthened,

They know him really well now because they've seen him a few times and obviously, they look after the whole family [talking about GP]. (P26)

In another example, one parent reflects on their interactions with a community health visitor, describing her as her 'rock' and a 'really good friend' as she provided support and coordination, arranging 'appointments' and 'chasing things up' during her many visits (P23). Unfortunately, relationship continuity was not always a shared experience for all as some would not see 'assigned' GP (P02) or there would be limited community services available in their area.

### Theme 3: equipment barriers

In attempts to source necessary equipment for their child, many parents found themselves facing several barriers such as inappropriate equipment, a lack of or lengthy waiting times. The NHS funding streams and contracts with providers seemed to be a contributor as parents felt that the standards of equipment were dependent on availability and capacity of the contracted company,

Equipment is purchased from particular providers because that's who the contract is with …. Doesn't necessarily mean that you get the best equipment for your child, it means you get the best equipment that particular contracted service can offer. (P18)

Conflict between parents and contracted equipment companies would arise as both had differing opinions on the standard of equipment provided. On the one hand, parents described it as a 'fight' to get 'a basic standard of living' whereas companies viewed items as 'luxury' (P14&15) and, therefore, it would not be provided under their current offer. Similar to NHS-provided equipment,

local councils are also set-up to help. However, these services were also sometimes limited in what they could provide. For unknown reasons, one family had not yet received funding from their council for a cot, leaving them to rely on charity resources in the meantime.

The council have basically said they've not got a bed for him and they're not willing to fund it yet … They've got a normal bed which we can have but that's not appropriate for him … A charity called [name] who loan us a cot for 6 months … Hopefully by that the time that loan comes to an end we'll have to come to some form of agreement to supply us with something that is appropriate. (P10&11)

If successful in sourcing equipment, waiting times would be long, some taking up to '6 weeks' (P01) and described as taking 'forever' (P03&04). There were also concerns relating to medication pick-up services as this was dependent on specific areas,

The hospital are really meant to supply us all his equipment and his consumable and syringes … It's really not practical for us because he takes up a lot of space in the care and all his equipment. (P10&11)

If it were, it was not always consistent,

I trialed the delivery … sometimes it just doesn't turn up and I can't afford for that to happen. (P24)

Another source of tension related to the mobility service provided by Disability Living Allowance (DLA) funded by the government as it was only available to those aged 3 years or older. This meant that many of the parents were not entitled to a mobility car or an allowance despite their child's condition being considered as degenerative,

We've fought from birth because they're never going to get better, their condition is degenerative [discussing inability to get a mobility car]. (P02)

The amount of equipment that comes with [child's name] and has to go with him everywhere because it's life-saving equipment, it's to me, horrific and atrocious that why has he got to be 3, do you suddenly have more of a mobility need after 3? (P10&11)

The different streams of equipment resources and tensions illustrate a larger problem which is widespread with parents experiencing implications at a variety of different avenues such as NHS, council or government.

### Theme 4: charities fill the gaps

Due to the complexity and additional needs for this patient group, it was not uncommon for families to face financial concerns, a result of long hospital stays, changes

to their work arrangements and other costs, leaving many to rely on charities for essential equipment, loans and respite. In one case, a parent was told that she would no longer be able to work due to her son's caregiving needs. However, the loss of income raised concerns and anxieties as it would not be feasible, leaving charities to fill the gaps.

> It was give up work originally, to which we then were quite fearful of what are we going to do? How are we going to afford everything? How are we going to be able to pay for our house, pay for living? (P03&04)

To care for her child, another parent also reduced her working hours to 1 day a week resulting in a significant decrease to their wages, causing financial concerns and reliance on charity support

> I do 1 day but I used to do full-time but obviously, I took a massive, massive cut on my wages (P13)

Financial concerns faced by families appeared to be a shared experience among this patient group as it was common practice for healthcare or social care professionals to refer to charities,

> There's a family worker at the hospital, so she helps us with … Anything to do with like charities that can help us. (P23)

Charities provided respite, granting parents an opportunity to have quality time with each other or some form of break from their caregiving duties,

> I think it was one of the nurses mentioned it and I looked into it. I Googled it after they'd mentioned "You can get holidays or equipment that you need" (P13)

It was also common for charities to provide equipment such as car seats and vehicles.

> I also got a car seat through a charity, which was an absolute godsend because I had to stop working as well (P13)

For many, it appeared that charities would close gaps surrounding equipment and support as social and healthcare services were not always able to successfully do this. Describing her interactions with social services, a parent felt as though their situation was misunderstood as she did not have her house furnished prior to her child's discharge. Despite experiencing a difficult journey impacting their circumstances, the parent felt judged and opted to use a charity service instead.

> They [social worker] didn't understand. [charity name] did understand. They helped ease off some

expenditure by providing white goods and that children's charity, they helped because they gave £500 Ikea vouchers. (P19)

In another case, charities acted as a last resort as the council was unable to provide a family with necessary equipment, leaving parents to feel frustrated.

> The frustrating part is obviously the council part of it, the fact they haven't got a bed and we're having to go through a charity. (P10&11)

Despite meaningful interactions and support from charities, the application process was complex and would frequently involve long waiting times to find out if their application was successful.

> We got told to go to this charity, which we have done, but we ordered the car in January and we have been told we probably won't get it until July. At the earliest. (P03&04)

There were also instances of when parents' application was unsuccessful, leaving families to source another charity before they were granted support.

> We had two different application process and the first one said, "You're not having it." And then it was [charity name] that said, "Yes, you can have it. (P01)

## DISCUSSION

This study explored parents' experiences of receiving care for their child with medical complexity with findings showing that parents feel abandoned and frequently not supported by services. Sourcing equipment was another area of conflict as it was unavailable or involved extensive processes, leaving parents to rely on charities to meet basic needs of their child. When present, parents greatly valued continuity of care, involving meaningful relationships with professionals, coordination and effective communication.

The published literature surrounding the role of a key worker, a coordinator of care for the wider population of children with disabilities and their families has existed since the late 1980s.[21] Given the increase in complexity and needs of this population of children, the role of coordinator is likely to be more complex. Recent national (NICE) guidance[22] recommended that future research should involve examining the effectiveness of 'dedicated key workers' and 'care closer to home'. Results from a recent randomised-controlled trial of care coordination for CMC in Canada[23] have shown that it improves perceptions, but not of other outcomes.

Studies from the USA have shown that parents value support from multifaceted interventions, which are designed to provide 'emotional, informational,

instrumental and financial support'.[24] Our study has shown that there continues to be inadequate support in relation to financial, informational and psychological needs. In line with previous research which found that almost one-fifth of parents had reported experiencing poor or fair mental health and were thought to be a high-risk group,[25] we found that many parents experienced psychological challenges, stemming from their child's diagnosis as they were provided with limited information, frequently relying on internet sources or other families, continuing as they moved through services. Financial support was found to be another area requiring revision as many relied on charities to access equipment such as car seats or wheelchairs, unavailable through NHS-provided services and were too expensive to buy privately. Issues with access to equipment and the importance of this to families of children with disabilities were first raised more than 20 years ago.[26] This was also found in another paper exploring the parental experiences of children using long-term ventilation as families faced struggles navigating a fragmented system.[27] It was seen as common practice for services to refer parents to charities, posing a cause for concern as many are further impacted by the cost of living crisis.[28]

Parents played an active role in their child's care, describing it as a fight. It is evident that parents play an important role in their child's care with recent calls[29] for them to be considered eligible for COVID-19 vaccinations alongside healthcare professionals. In this study, the importance of their role was illustrated through their training abilities and staff dependency when short-staffed, preventing rest. Concerns surrounding staff shortages and absences from COVID-19 have been found in recent UK reports[30] as hospitals remain under pressure. Parents experiencing challenges when navigating their child's care are supported in other research.[31] To help lessen such challenges, parents also reflected on positive instances of care, valuing familiar relationships. This is supported in other research[11] as the importance of familiar team members, leaving parents feeling reassured and comforted knowing that staff members were aware of their child and routine, has been emphasised. Unfortunately, relationship continuity was identified as inconsistent due to staffing ratios in accident and emergency departments, for example. However, due to the limited expertise in their community, preference of care locations was limited. The acute environment itself was viewed as inappropriate due to lengthy waiting times, another shared experience[13] among this patient cohort.

### Implications for services

In this study, parents made recommendations for future services, involving increased expertise in the community and local hospital sites. There were examples of when dedicated complex care services were beneficial, strengthening future implementation. It is understood that given limited resources and current pressures faced by health services, implementation may be difficult. However,

efforts to better support parents from the period of diagnosis and as they move through the health service should be made. It may also be useful to use existing children's palliative care teams as they possess the relevant skills to effectively care for this population. Positive instances relating to continuity of care were found and may be one suggested way of achieving this. Government assistance, for example, DLA should be re-examined to fit the needs of this population. If referral to charity services is becoming standard practice, guidance and changes to paperwork to help simplify applications should be considered.

### Implications for research

There is an urgent need to evaluate interventions which promote integration of care across community, hospital and specialist teams for these children and their families. Future research should also explore how cultural and socioeconomic factors may impact parents' experiences in receiving care for their child.

### Strengths and limitations

The semi-structured nature of the interviews allowed parents to share their narratives, producing meaningful and rich data. The research was also collected during waves of the COVID-19 pandemic, meaning that unique lived experiences during this time were captured. Participants were also recruited from multiple sites with different geographical locations, some hospitals with dedicated complex services and some without. There are some limitations in this study. Due to funding constraints, only English-speaking participants were recruited. Therefore, the opportunity to interview other parents could have been missed. We did not report ethnicity and were also unable to explore cultural and socioeconomic factors that may influence parents' experiences, meaning that important insights may have been lost. Unfortunately, we were unable to recruit any parents from one of the NHS sites due to unknown reasons, meaning that some perspectives may not have been captured. By using purposeful sampling to recruit parents using the recommendations of clinicians, there is the risk that selection bias may have occurred and therefore, findings may not be representative of all parents. Like other qualitative research studies, there is a risk of researcher bias as their background and perspectives may influence this. However, in this study, a series of three different researchers with different backgrounds and experiences were involved in the debriefing and analysis process. It is also understood that due to most of the existing literature being US based, it may mean that some findings from this study may not be applicable to other healthcare contexts.

### CONCLUSIONS

Parents of CMC experienced many challenges when receiving care for their children; many of these challenges have been highlighted over the past two to three

decades but despite the children's needs becoming more complex, little progress appears to have been made. Parents were seen as adopting significant additional roles, beyond being a parent in their child's life but they still find themselves left without support across all areas of their child's life. Families of CMC require more structured support, policy makers and commissioners need to prioritise the needs of these families to enable health and social care services to provide the support required.

**Acknowledgements** The authors would like to thank members of the Martin House Centre's family advisory board and CoLab for their valuable contributions during the study design phase. They would like to thank the clinicians acting as a principal investigator at hospital sites for all of their help. They would also like to thank all parents involved in this research for sharing their story.

**Contributors** LKF designed this study and secured funding. EVM collected and analysed the data with regular input from LKF and JH. EVM drafted the manuscript, both LKF and JH made considerable contributions.

**Funding** This study was funded by the Children's Hospital Alliance (grant number: N/A). The National Institute for Health Research (NIHR) Career Development Fellowship (Award CDF-2018-11-ST2-002) was awarded to LF and contributed to this research project.

**Disclaimer** All opinions expressed in this paper are strictly the authors. LF (acting as the guarantor and is responsible for the overall content of this study) led the design of this study, wrote the protocol and secured funding with valuable input from CoLab. Members of the Martin House Research Centre's Family Advisory Board helped to shape the topic guides, provided feedback on inital findings and assisted with a mock interview to trial the topic guides. EVM collected the data (overseen by LF) and later analysed with LF and JH to determine themes. Themes were discussed between all authors (LF, EVM & JH) when drafting the final report. All authors made a considerable contribution to the manuscript drafts as EVM drafted the original manuscript, LF and JH provided comments and changes until a finalised version was agreed upon.

**Competing interests** None declared.

**Patient and public involvement** Patients and/or the public were involved in the design, or conduct, or reporting, or dissemination plans of this research. Refer to the Methods section for further details.

**Patient consent for publication** Not applicable.

**Ethics approval** This study involves human participants and ethical and governance approvals were obtained from HRA and Health and Care Research Wales (08/09/21, 300516) and Research Ethics Committee (08/09/21, 21/WA/0257). Participants gave informed consent to participate in the study before taking part.

**Provenance and peer review** Not commissioned; externally peer reviewed.

**Data availability statement** All data relevant to the study are included in the article or uploaded as supplementary information.

**ORCID iD**
Emma Victoria McLorie http://orcid.org/0000-0003-2043-7069

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
