## [Reviewer comments · BMJ Paediatrics Open]

ARTICLE DETAILS

TITLE (PROVISIONAL)	Understanding Parents' Experiences of Care for Children with Medical Complexity in England: A Qualitative Study
AUTHORS	McLorie, Emma Fraser, Lorna Hackett, Julia

VERSION 1 - REVIEW

REVIEWER	Dr. Simon Lenton
REVIEW RETURNED	05-Jun-2023

GENERAL COMMENTS	This is a well written and easily understood paper. I cannot comment whether reflexive thematic analysis was the correct qualitative research tool to use with these interviews. Like many qualitative research studies there is potentially a problem with selection bias since parents were not chosen randomly, but rather through the recommendations of clinicians and by volunteering through social media, this should be acknowledged in the discussion. However, the themes identified are congruent with previous research publications and my experience of services. It is incredibly disappointing that services appear not to have significantly improved over the last 20 years. I think the Margolan paper entitled "Parental experience of services when their child requires long-term ventilation" came to very similar conclusions in 2004. It would be useful to reference good practice, for example the Barnardos paper "from hospital to home guidance on discharge management and community support for children using long-term ventilation" (2005) being one of the earlier examples. There is a numeral 4 in table 3 without text. One of the recommendations for future services is that there should be increased expertise in the community and within local hospital. Some reference to children's palliative care teams might be helpful here as they should have the necessary skills and competencies to manage technology dependent children in the later stages of their life. The terms community, hospital and specialist teams are better than primary, secondary and tertiary health care in an era of increasingly integrated networked services. The references are somewhat chaotic with text and numbers missing for example references 3, 7, 12, 17, 22.
--

REVIEWER	Dr. Surendra Gupta Childrens Medical Center of Fresno, Pediatrics
REVIEW RETURNED	05-Jun-2023

GENERAL COMMENTS

Overall, the study provides valuable insights into the experiences of parents of children with Medical Complexity in the UK healthcare system. It addresses the gap in research regarding the specific needs and challenges faced by this population and their families. The study's methodology appears to be appropriate for the research question and objectives. The findings highlight the need for improved support for parents and children with Medical Complexity, including increased expertise in community and local hospital settings, continuity of care, and changes to government assistance programs.

It is important to consider following suggestion and limitations when interpreting the findings of the study and to recognize the need for further research to address these limitations and provide a more comprehensive understanding of the experiences and needs of parents and children with medical complexity.

Suggested title for the study:

Title: Understanding Parents' Experiences of Care for Children with Medical Complexity in England: A Qualitative Study

Limitations in the study:

Limited generalizability: The study acknowledges that much of the existing literature on models of care for children with medical complexity is based on research conducted in North America., where healthcare systems differ significantly from the UK's publicly funded system. Therefore, the generalizability of research findings from the US to the UK population is questionable. This limitation reduces the applicability of the study's findings to other healthcare contexts

Sampling bias: The study included a relatively small sample size of 20 parents of CMC, which may limit the representativeness and generalizability of the findings. The study states that purposeful sampling was used across four tertiary hospitals in England, but it doesn't provide a clear rationale for the selection of these hospitals. Additionally, the study acknowledges that one site failed to recruit any participants, which may introduce bias and limit the representativeness of the sample.

Potential researcher bias: The study mentions that interviews were conducted by a female researcher who was experienced in interviewing. While this experience is valuable, the researcher's background and perspective may introduce bias in data collection, interpretation, and analysis. Efforts should be made to minimize bias and ensure objectivity in the research process.

Insufficient recommendations for future research: While the study mentions the need to evaluate different models of care and integration across healthcare levels, it doesn't provide specific recommendations for future research. Elaborating on the research gaps and potential avenues for further investigation would enhance the study's contribution to the field.

Limited exploration of cultural and socioeconomic factors: The study does not explicitly explore how cultural or socioeconomic factors may influence parents' experiences and access to care for their children with medical complexity. Considering these factors could provide important insights into the specific challenges faced by different populations and inform more targeted interventions and support services.

VERSION 1 – AUTHOR RESPONSE

Reviewer: 1

This is a well written and easily understood paper. I cannot comment whether reflexive thematic analysis was the correct qualitative research tool to use with these interviews. Like many qualitative research studies there is potentially a problem with selection bias since parents were not chosen randomly, but rather through the recommendations of clinicians and by volunteering through social media, this should be acknowledged in the discussion.

Response: Thank you for this helpful comment. I have added this in the discussion section of the paper under the '*Strengths and limitations*' section. See text here: "*By using purposeful sampling to recruit parents using the recommendations of clinicians, there is the risk that selection bias may have occurred and therefore, findings may not be representative of all parents*"

However, the themes identified are congruent with previous research publications and my experience of services. It is incredibly disappointing that services appear not to have significantly improved over the last 20 years. I think the Margolan paper entitled "Parental experience of services when their child requires long-term ventilation" came to very similar conclusions in 2004.

Response: Thank you for bringing this to my attention, I really appreciate it. I've added this reference into the discussion section. See text here: "*This was also found in another paper exploring the parental experiences of children using long-term ventilation as families faced struggles navigating a fragmented system*" found on page 20.

It would be useful to reference good practice, for example the Barnardos paper "from hospital to home guidance on discharge management and community support for children using long-term ventilation" (2005) being one of the earlier examples.

Response: Thank you for this. I've added this reference into the introduction of the paper found on page 4. See text here: "*Although a previous UK-based report provided guidance on transitional care from the perspective of children using long-term ventilation, illustrating their complex needs*"

There is a numeral 4 in table 3 without text.

Response: Thank you for spotting this. I've removed it as seen on page 9.

One of the recommendations for future services is that there should be increased expertise in the community and within local hospital. Some reference to children's palliative care teams might be helpful here as they should have the necessary skills and competencies to manage technology dependent children in the later stages of their life.

Response: Thank you – I've added this under the '*Implications for services*' section found on page 20. See text here: "*It may also be useful to utilise existing children's palliative care teams as they possess the relevant skills to effectively care for this population*"

The terms community, hospital and specialist teams are better than primary, secondary and tertiary health care in an era of increasingly integrated networked services.

Response: Thank you for this useful feedback, I've changed these instances found under the '*Implications for research*' section found on page 21. See text here: "*There is an urgent need to evaluate interventions which promote integration of care across community, hospital, and specialist teams for these children and their families*"

The references are somewhat chaotic with text and numbers missing for example references 3, 7, 12, 17, 22.

Response: Apologies, there had been some difficulties with Endnote. I've since addressed this and made changes, ensuring that all references are correct and up to date. Thank you again.

Reviewer: 2

Overall, the study provides valuable insights into the experiences of parents of children with Medical Complexity in the UK healthcare system. It addresses the gap in research regarding the specific needs and challenges faced by this population and their families. The study's methodology appears to be appropriate for the research question and objectives. The findings highlight the need for improved support for parents and children with Medical Complexity, including increased expertise in community and local hospital settings, continuity of care, and changes to government assistance programs.

It is important to consider following suggestion and limitations when interpreting the findings of the study and to recognize the need for further research to address these limitations and provide a more comprehensive understanding of the experiences and needs of parents and children with medical complexity.

Response: Thank you for this, we completely agree. We've addressed this and added in a more detailed '*strength and limitations*' section found on page 21 in the discussion using guidance from Reviewer 1 and your comments below. Further details and exact text can be found below.

Suggested title for the study:

Title: Understanding Parents' Experiences of Care for Children with Medical Complexity in England: A Qualitative Study

Response: Thank you for this useful suggestion – we think it works really well. We've now changed the title to: "*Understanding Parents' Experiences of Care for Children with Medical Complexity in England: A Qualitative Study*" as found on Page 1.

Limitations in the study:

Limited generalizability: The study acknowledges that much of the existing literature on models of care for children with medical complexity is based on research conducted in North America., where healthcare systems differ significantly from the UK's publicly funded system. Therefore, the generalizability of research findings from the US to the UK population is questionable. This limitation reduces the applicability of the study's findings to other healthcare contexts

Response: Thank you, we agree with this and have added this in the new section, '*Strengths and Limitations*' as found on page 21. See text here: "*It is also understood that due to most of the existing literature being US based, it may mean that some findings from this study may not be applicable to other healthcare contexts*"

Sampling bias: The study included a relatively small sample size of 20 parents of CMC, which may limit the representativeness and generalizability of the findings.

Response: Thank you for your comment and useful feedback.

We believe that our sample size is effective and suitable for this study as data saturation was reached using guidance from Braun and Clarke. In addition to this, Braun and Clarke have further commented on sample size, warning researchers' of potential pitfalls of a large sample size.

See here: *“With an organic and flexible approach to TA, and a very wide range of potential project sizes, and data sources, it is expected and appropriate that samples would vary considerably in size. Moreover, if we do not conceptualise themes as diamonds waiting to be discovered, we don’t have to rely on the idea of a truth we might miss – and hence do not need to chase the relatively large sample sizes (for interview-based qualitative research) that Fugard and Potts’ model produces. Bigger isn’t necessarily better. The bigger the sample, the greater the risk of failing to do justice to the complexity and nuance contained within the data.”* (Braun and Clarke, 2015) – Titled: (Mis)conceptualising themes, thematic analysis, and other problems with Fugard and Potts’ (2015) sample-size tool for thematic analysis Virginia Braun and Victoria Clarke Commentary for International Journal of Social Research Methodology.

The study states that purposeful sampling was used across four tertiary hospitals in England, but it doesn’t provide a clear rationale for the selection of these hospitals.

Response: Thank you for spotting this, really appreciate it. I’ve now added more context on Page 5 under the *‘design and participants’* section found on Page 5. See text here: *“These hospital sites were chosen as each were known to treat children with medical complexity. The three hospitals which did have existing services also differed from one another, some with more established teams whereas others were relatively new and involved a single individual. Therefore, it was likely that experiences may differ depending on type of service received”*

Additionally, the study acknowledges that one site failed to recruit any participants, which may introduce bias and limit the representativeness of the sample.

Response: Thank you for this, this has been removed from the previous *‘strengths and limitation’* bullet point section and now in the discussion found on page 21. See text here *“Unfortunately, we were unable to recruit any parents from one of the NHS sites due to unknown reasons meaning that some perspectives may not have been captured”*

Potential researcher bias: The study mentions that interviews were conducted by a female researcher who was experienced in interviewing. While this experience is valuable, the researcher’s background and perspective may introduce bias in data collection, interpretation, and analysis. Efforts should be made to minimize bias and ensure objectivity in the research process.

Response: Thank you for raising this, we completely agree. We feel that some of this was addressed in our data analysis section as fieldnotes were taken. Two other team members including EVM with a range of different backgrounds were involved in the data analysis process (EVM, LF and JH) as noted in the data analysis section.

However, I’ve now made changes to make this clearer in three parts of the paper. See below:

1. Strengths and limitation section on page 21 – See text: *“By using purposeful sampling to recruit parents using the recommendations of clinicians, there is the risk that selection bias may have occurred and therefore, findings may not be representative of all parents. Like other qualitative research studies, there is a risk of researcher bias as their background and perspectives may influence this. However, in this study, a series of three different researchers with different backgrounds and experiences were involved in the debriefing and analysis process”*
2. Data collection section found on page 5 – See text: *“During the interview, field notes were taken to reduce researcher bias”*
3. Data collection section found on pages 5-6 See text: *“After interviews were conducted by EVM, debriefs were held with LF or JH which also allowed opportunity to discuss any potential thoughts or potential bias”*

Insufficient recommendations for future research:

While the study mentions the need to evaluate different models of care and integration across healthcare levels, it doesn't provide specific recommendations for future research. Elaborating on the research gaps and potential avenues for further investigation would enhance the study's contribution to the field.

Limited exploration of cultural and socioeconomic factors: The study does not explicitly explore how cultural or socioeconomic factors may influence parents' experiences and access to care for their children with medical complexity. Considering these factors could provide important insights into the specific challenges faced by different populations and inform more targeted interventions and support services.

Response: Thank you for your useful feedback and bringing this to our attention, I really appreciate it. We've addressed this in two sections of the paper. See below:

- Implications for research section found on page 21. See text: *Future research should also explore how cultural and socioeconomic factors may impact parents' experiences in receiving care for their children with medical complexity*". We also removed the following text from this section "*Due to the high use of emergency and accident departments found and in hopes of reducing unnecessary utilization, we posit that there is scope to evaluate community services and explore clinicians' experiences when caring for this population.*" as we feel that it did not provide insight into research gaps/future research.
- Strengths and limitations section found on page 21. See text: "*Due to funding constraints, only English-speaking participants were recruited. Therefore, the opportunity to interview other parents could have been missed. We did not report ethnicity and were also unable to explore cultural and socioeconomic factors that may influence parents' experiences meaning that important insights may have been lost. Unfortunately, we were unable to recruit any parents from one of the NHS sites due to unknown reasons meaning that some perspectives may not have been captured.*"